# Advances and Remaining Challenges in the Treatment for Borderline Resectable and Locally Advanced Pancreatic Ductal Adenocarcinoma

**DOI:** 10.3390/jcm11164866

**Published:** 2022-08-19

**Authors:** Megan L. Sulciner, Stanley W. Ashley, George Molina

**Affiliations:** 1Department of Surgery, Brigham and Women’s Hospital, Boston, MA 02115, USA; 2Dana–Farber/Brigham and Women’s Cancer Center, Boston, MA 02215, USA

**Keywords:** borderline resectable, locally advanced, pancreatic ductal adenocarcinoma, neoadjuvant therapy, cancer care disparities

## Abstract

Pancreatic ductal adenocarcinoma (PDAC) remains one of the deadliest malignancies in the United States. Improvements in imaging have permitted the categorization of patients according to radiologic involvement of surrounding vasculature, i.e., upfront resectable, borderline resectable, and locally advanced disease, and this, in turn, has influenced the sequence of chemotherapy, surgery, and radiation therapy. Though surgical resection remains the only curative treatment option, recent studies have shown improved overall survival with neoadjuvant chemotherapy, especially among patients with borderline resectable/locally advanced disease. The role of radiologic imaging after neoadjuvant therapy and the potential benefit of adjuvant therapy for borderline resectable and locally advanced disease remain areas of ongoing investigation. The advances made in the treatment of patients with borderline resectable/locally advanced disease are promising, yet disparities in access to cancer care persist. This review highlights the significant advances that have been made in the treatment of borderline resectable and locally advanced PDAC, while also calling attention to the remaining challenges.

## 1. Introduction

The incidence of pancreatic ductal adenocarcinoma (PDAC) has more than doubled in the last three decades [1]. Despite overall survival at 5 years remaining low at approximately 5% [2], there have been significant advances in the treatment of borderline resectable and locally advanced disease. The addition of neoadjuvant chemotherapy has been associated with improved overall survival by downstaging disease and providing more patients with the opportunity to proceed to surgical resection, which remains the only curative option for PDAC. Yet, notably, the determination of whether the disease has been downstaged by radiographic modalities remains unclear. Although adjuvant treatment improves survival in patients who have received upfront surgical resection, the benefit of adjuvant therapy in patients who have received neoadjuvant therapy is not as clear. Continued progress in the treatment of patients with PDAC will depend largely on clarifying the optimal neoadjuvant and adjuvant treatments for patients with borderline resectable and locally advanced disease. Additionally, more work needs to be done to address disparities in access to cancer care for all patients with PDAC.

## 2. Radiographic Classification

PDAC at time of diagnosis is categorized by the radiographic distribution of disease and the relationship between the tumor and the surrounding vasculature. Historically, tumors have been termed either resectable or non-resectable based on the prediction of achieving negative surgical margins. However, new additional subgroups have emerged based on vascular involvement termed borderline resectable and locally advanced disease. Borderline resectable and locally advanced tumors reside on the continuum between resectable and non-resectable. Approximately 30% of all patients with PDAC are characterized as having borderline resectable or locally advanced disease at the time of diagnosis. However, the classification of borderline resectable and locally advanced PDAC has evolved and at present there is not a consistently accepted definition. 

At our institution, Dana–Farber/Brigham Cancer Center (DF/BCC), we have incorporated the definitions from the Americas Hepato-Pancreato-Biliary Association/Society of Surgical Oncology/Society for Surgery of Alimentary Tract (AHPBA/SSAT/SSO), MD Anderson Cancer Center (MDACC), and the National Comprehensive Cancer Network (NCCN) in combination with our experience in treating patients with PDAC (Table 1) [3,4,5]. We define borderline resectable as including tumors that either abut, encase, or occlude a reconstructable portion of the superior mesenteric vein (SMV)/portal vein (PV). Borderline resectable tumors can abut the superior mesenteric artery (SMA), celiac trunk, and common hepatic artery. Tumors that occlude a reconstructable portion of the common hepatic artery are also included in our definition of borderline resectable. Our definition of locally advanced disease includes tumors that occlude a portion of the SMV/PV that cannot safely be reconstructed or that encase the SMA, celiac trunk, or common hepatic artery. These definitions guide our clinical decision making about the sequence of chemotherapy and surgery (i.e., if curative-intent surgical resection is possible) and whether radiation is included in the treatment regimen.

## 3. Neoadjuvant Chemotherapy

Using national population-level data, neoadjuvant therapy has been found to be associated with an overall survival benefit in patients with borderline resectable and locally advanced PDAC. The rationale for treating patients with neoadjuvant therapy stems from the anticipated potential of tumor downstaging [6]. Downstaging improves the chance of tumor resectability and decreases the risk of positive surgical margins [6,7]. In a study using the National Cancer Database (NCDB) that included nearly 2000 patients with borderline resectable pancreatic cancer who were treated with neoadjuvant therapy between 2004 and 2015, the median overall survival was similar to patients with upfront resectable disease [8]. Additionally, among patients with borderline resectable disease, neoadjuvant therapy was associated with an improved overall survival compared to adjuvant therapy. Patients who received neoadjuvant therapy were less likely to have positive surgical margins [8], highlighting the possible role of neoadjuvant therapy in downstaging disease.

Several single-institutional series have shown promising results with neoadjuvant FOLFIRINOX contributing to successful R0 resection and improved overall survival among patients with borderline resectable/locally advanced PDAC. A retrospective study by Ferrone and colleagues that included 40 patients with either borderline resectable or locally advanced disease and who were all treated with neoadjuvant FOLFIRINOX reported that 92% of patients underwent an R0 resection [9]. A follow-up study that included 141 patients with borderline resectable/locally advanced diseases that were surgically explored following treatment with neoadjuvant FOLFIRINOX reported that 78% of patients underwent surgical resection [10]. This latter study has reported the best median overall survival thus far in the literature—37.7 months for all patients who were resected and 34.2 months among all patients who received neoadjuvant FOLFIRINOX. A meta-analysis of 24 studies reporting on outcomes following neoadjuvant FOLFIRINOX treatment reported an overall patient-level median survival of 22.2 months [11]. Since resection with negative margins is the only known curative treatment for PDAC, neoadjuvant FOLFIRINOX should be considered in patients with borderline resectable/locally advanced disease as it is associated with an improved likelihood of undergoing an R0 surgical resection and potentially improved survival.

However, clinical trials to date have reported mixed evidence in support of neoadjuvant chemotherapy in the treatment for PDAC. The PREOPANC trial was the first prospective, randomized trial comparing neoadjuvant therapy and upfront resection for borderline resectable PDAC [12]. This multi-center, phase 2/3 trial reported a higher rate of R0 resection (51.8% versus 26.1%) and improved median overall survival (21 months versus 12 months) for patients with borderline resectable PDAC who received neoadjuvant chemoradiotherapy compared to patients who underwent upfront surgery followed by adjuvant chemoradiation using gemcitabine as the chemotherapy [12]. A limitation to this study is that the treatment regimen included single-agent gemcitabine as the chemotherapy backbone, as opposed to the more commonly used FOLFIRINOX or gemcitabine with nab-paclitaxel. The FOLFIRINOX regimen includes oxaliplatin (85 mg/m^2^), irinotecan (180 mg/m^2^), and 5-fluorouracil (2400 mg/m^2^) administered every 2 weeks for a duration of 2 months. Subsequently, patients are restaged and then provided with an additional 2 months of FOLFIRINOX therapy [13]. If patients do not demonstrate disease progression at restaging and are deemed candidates for surgical resection, they proceed to surgery. Alternatively, gemcitabine/nab-paclitaxel includes nab-paclitaxel (125 mg/m^2^) followed by gemcitabine (1000 mg/m^2^). This regimen is provided every week for 3 weeks followed by 1 week without chemotherapy for a total of 2 months (Table 2). Patients are then restaged and receive an additional 2 months of chemotherapy with plans to proceed to surgery if there is no disease progression and they are deemed candidates for surgical resection at restaging.

Murphy and colleagues reported an R0 resection rate of 65% among all patients who were treated with neoadjuvant FOLFIRINOX in a single-arm, phase II clinical trial. Among the 32 out of 48 patients that underwent a curative-intent surgical resection, 97% had a R0 resection. Median overall survival was 37.7 months for all patients. Although the median overall survival had not been reached at time of publication for the cohort of patients that underwent surgical resection, the progression median survival was 48.6 months, and the 2-year overall survival was 72% [14].

In another phase II clinical trial that compared neoadjuvant-modified FOLFIRINOX with or without radiation therapy, patients treated without radiation had a median overall survival of 31 months with a median follow-up of 27 months. The only randomized controlled trial that compared modified FOLFIRINOX and gemcitabine/nab-paclitaxel in the neoadjuvant setting was performed on patients with radiographic resectable disease [14]. The SWOG-S1505 study, a randomized phase 2 trial that included 103 patients with resectable PDAC, found no survival difference between neoadjuvant-modified FOLFIRINOX and gemcitabine/nab-paclitaxel [13]. Unfortunately, the median overall survival for each neoadjuvant regimen was approximately 23 months, equivalent to the historical overall survival [13]. Nevertheless, neoadjuvant chemotherapy was well-tolerated, and 95% of patients successfully underwent surgical resection after neoadjuvant chemotherapy. Notably, 85% of patients had negative margins, and 33% of patients had a major pathologic response after neoadjuvant chemotherapy [13]. Thus, while neoadjuvant chemotherapy does appear to improve the rate of resectability and is well-tolerated, the overall survival benefit thus far has been based on single-institution studies and remains an area of ongoing investigation. Randomized controlled trials are needed to ultimately determine the utility of neoadjuvant chemotherapy for patients with borderline resectable/locally advanced disease.

## 4. Imaging after Neoadjuvant Therapy for Downstaging

Restaging imaging after neoadjuvant therapy for borderline resectable or locally advanced PDAC is performed to determine if the tumor has been downsized and subsequently has become more amenable to surgical resection. Accuracy in predicting the resectability of PDAC of the head of the pancreas has been shown to decrease after neoadjuvant chemotherapy [15]. Additionally, while neoadjuvant FOLFIRINOX has been found to be associated with decreased disease burden on CT imaging among patients with borderline resectable/locally advanced disease, CT imaging cannot predict resectability or pathologic response [16]. In a study of 129 patients with borderline resectable disease who received neoadjuvant therapy, downstaging after neoadjuvant therapy based on CT imaging was rare (0.8% of all patients) [17]. Given the inability to determine downstaging on CT, the authors of this study recommended patients undergo pancreatectomy after neoadjuvant therapy if there is no evidence of metastases [17]. In a retrospective blinded review of preoperative imaging of either borderline resectable or locally advanced disease in patients who received neoadjuvant FOLFIRINOX, 32% were considered to be unresectable, 56% were considered borderline resectable, and only 12% were considered resectable. All patients proceeded to the operating room and 27% of patients were found to be unresectable, while 70% achieved negative surgical margins [18]. Thus, these findings demonstrate that, while CT has acceptable sensitivity, it has low specificity in predicting resectability after neoadjuvant FOLFIRINOX [18]. One explanation for the discrepancy in imaging and operative findings for borderline resectable/locally advanced patients who have received neoadjuvant chemotherapy is the dissolution of tissue planes secondary to cytotoxic treatment, resulting in fibrosis which is difficult to discern from malignancy on imaging [15,18]. The determination of surgical candidacy after neoadjuvant therapy for borderline resectable/locally advanced patients should not be based on imaging findings alone. We recommend surgical exploration for patients who have a treatment response or stable disease on restaging scan following the completion of neoadjuvant treatment.

## 5. Neoadjuvant Therapy May Also Be Beneficial for Upfront Resectable PDAC

Surgical resection remains the only curative treatment option for PDAC if negative margins are achieved. However, even with an R0 resection, many patients will develop recurrence. Several studies including a phase 3 clinical trial have demonstrated an overall survival benefit with neoadjuvant chemotherapy as compared to upfront resection for early-stage/upfront resectable PDAC [19,20,21]. The results from the Alliance A021806 phase III trial that is comparing perioperative versus adjuvant chemotherapy for resectable PDAC will provide needed clarity about the role of neoadjuvant chemotherapy in patients with upfront resectable disease [22].

## 6. Adjuvant Chemotherapy

The role of adjuvant therapy for patients with borderline resectable/locally advanced disease who received neoadjuvant therapy is unclear. A retrospective study by Perri and colleagues that included 245 patients with PDAC who received neoadjuvant chemotherapy reported that patients who received adjuvant chemotherapy had increased overall survival (42 versus 32 months) and recurrence-free survival (17 versus 12 months) compared to patients who did not receive adjuvant therapy [23]. In an international multi-center retrospective study among patients who received neoadjuvant FOLFIRINOX and underwent surgical resection, adjuvant chemotherapy was associated with a higher overall survival only in patients who had node-positive disease on pathology evaluation [24]. Similarly, in a retrospective NCDB study that included patients with borderline resectable or locally advanced disease who were treated with neoadjuvant therapy, the addition of adjuvant therapy was associated with a survival benefit only among patients with positive margins after resection [25]. Among patients with stage I-II disease, another NCDB study found that adjuvant therapy after neoadjuvant therapy was associated with higher overall survival only among patients with low-risk pathologic features (i.e., low lymph node ratio, low-grade histology, and negative margin status) [26]. Taken together, these studies highlight the potential benefit of adjuvant therapy. However, the specific patient population with borderline resectable/locally advanced PDAC that may benefit has yet to be defined.

The highest overall survival reported thus far in the literature among all patients with PDAC has been in patients who underwent surgical resection and received adjuvant chemotherapy. In the PRODIGE-24 prospective trial, patients with resected PDAC were randomized to receive either adjuvant modified FOLFIRINOX or gemcitabine [27]. This study found that the median disease-free survival was increased in the modified FOLFIRINOX group to 21.6 months, as compared to 12.8 months in the gemcitabine group [28]. Additionally, the modified FOLFIRINOX group had an increased median overall survival of 54.5 months, compared to 35.0 months for the gemcitabine group [28]. However, notably, there was a higher rate of grade 3–4 adverse events with modified FOLFIRINOX (75.9% versus 52.9%) [28]. The overall survival benefit of adjuvant FOLFIRINOX that was reported in the PRODIGE-24 trial surpasses the results of the neoadjuvant studies [9,10,11].

## 7. Radiation Therapy

Radiation therapy as an adjunct to surgery aims to sterilize the tumor margins to ideally achieve negative margins at surgical resection. The timing of radiation in relation to resection has been explored. A recent NCDB study demonstrated that while there was no survival benefit with neoadjuvant chemoradiation compared to chemotherapy alone, patients who received neoadjuvant chemoradiation were more likely to have negative margins following resection [27]. Similarly, in a retrospective single-institution study, among patients who received neoadjuvant chemoradiation, R0 resection was achieved in 96% of patients with borderline resectable disease and in 88% of patients with locally advanced disease [29]. However, the patients in this study also had an overall locoregional failure rate of 33% after resection [29]. These studies highlight the potential benefit of neoadjuvant radiation in combination with chemotherapy. 

Individualized chemoradiation duration based on imaging of vascular involvement has demonstrated improved rates of R0 resection in a phase II clinical trial [14]. These patients all received neoadjuvant FOLFIRINOX, and if vascular involvement was found upon restaging imaging, these patients completed a longer course of chemoradiation as opposed to patients without vascular involvement [14]. Additionally, in patients with locally advanced disease, neoadjuvant chemoradiation may also be beneficial. Although there was no survival difference between additional chemotherapy versus chemoradiotherapy among patients with stable locally advanced disease on induction chemotherapy, significantly fewer patients who received chemoradiotherapy had disease progression (32% versus 46%) [30]. Additionally, the inclusion of chemoradiotherapy did not increase grade 3 or 4 toxicity [30]. In summary, the addition of chemoradiotherapy may be beneficial for patients with significant vascular involvement or locally advanced disease.

The Alliance A021501 trial, a prospective, multi-center clinical study, compared overall survival in patients with borderline resectable disease who were randomized to neoadjuvant modified FOLFIRINOX versus modified FOLFIRNOX plus radiation therapy (stereotactic body radiation therapy or hypofractionated radiation therapy) [31]. The addition of radiation therapy did not improve overall survival in patients with borderline resectable PDAC. For patients who underwent a pancreatectomy, 88% (28 out of 32) of patients who had neoadjuvant-modified FOLFIRINOX alone achieved an R0 resection compared to 74% (14 out of 19) of patients who had the addition of radiation therapy [31]. The two patients with a complete pathologic response were in the modified-FOLFIRINOX-plus-radiation arm. Median overall survival was 29.8 months (95% CI 21.1–36.6 months) for the modified-FOLFIRINOX-alone arm versus 17.1 months (12.8–24.4 months) in the modified-FOLFIRINOX-with-radiation arm [31]. Due to these findings, the authors concluded that the addition of radiation to modified FOLFIRINOX could be considered effective [31]. In our opinion, after completion of neoadjuvant chemotherapy (at our institution we prefer 4 months of FOLFIRINOX), we recommend the addition of radiation therapy prior to proceeding to surgery if there is deformation of the PV/SMV confluence or arterial involvement.

Another approach for improving local disease control is use of intraoperative radiotherapy, particularly for patients with borderline resectable or locally advanced disease. By providing precise, high-dose radiation directly to the tumor bed following resection, intraoperative radiation aims to reduce, or ideally eliminate, microscopic residual disease. In early studies, application of intraoperative electron beam radiation demonstrated marginal benefit when used alone for patients with unresectable disease [32]. However, in several small single-institution studies, the addition of intraoperative radiotherapy in resectable disease has demonstrated improved overall survival compared to surgery alone [33,34,35,36,37]. The addition of intraoperative radiotherapy for patients with borderline resectable or locally advanced disease with resection has demonstrated an improved progression-free survival of 21.5 months and overall survival of 46.7 months [34]. Similarly, Keane et al., demonstrated a median overall survival of 35.1 months in patients with borderline resectable/locally advanced disease who received resection and intraoperative radiotherapy compared to a median overall survival of 24.5 month for patients who had resection alone [35].

Additionally, for patients with locally advanced disease who received intraoperative radiotherapy combined with chemotherapy, median overall survival was 17.6 months, compared to 10.7 months for patient who received intraoperative radiation alone [36]. Interestingly, the median overall survival benefit was not dependent on the timing of chemotherapy, either preoperative or postoperative [36]. The goal of surgical resection is to achieve negative microscopic margins or an R0 resection. However, this is challenging especially in borderline resectable or locally advanced disease. In one of the largest single-institution studies inclusive of 201 patients with borderline resectable or locally advanced disease who received neoadjuvant chemoradiation and intraoperative radiotherapy, there was no disease-free or overall survival benefit with an R0 compared to R1 resection [37]. This finding supports the argument that intraoperative radiotherapy may aid in eliminating microscopic residual disease. While intraoperative radiotherapy may benefit patients with borderline resectable or locally advanced disease, the current studies evaluating its efficacy in this patient population are single institution studies. Therefore, these results are limited by small sample size and selection bias. Randomized control trials are needed to further define the patient population that could benefit from intraoperative radiotherapy.

## 8. Disparities in Access to Cancer Care for PDAC

While significant advances have been made in the treatment of borderline resectable and locally advanced PDAC, inequality in access to cancer care remains a reality in the United States. Patients have improved overall survival outcomes if surgical resection is achieved, yet not all patients in the United States undergo resection, regardless of resectability. Patients with PDAC were less likely to undergo surgical resection if they did not have a college education, had an annual median income less than or equal to the 25th percentile, were Black, were on Medicare or Medicaid, or were older than 65 years of age [38]. Black patients were also less likely to receive adjuvant chemotherapy or have consultation with an oncologic specialist, such as a radiation oncologist, medical oncologist, or surgical oncologist [39]. These findings were initially reported well over a decade ago, and yet inequality in access to cancer care for PDAC still persists.

In a retrospective study of 15,482 patients from the NCDB from 2004–2016, non-White patients were 25% less likely to undergo resection for borderline resectable PDAC as compared to White patients. Interestingly, if neoadjuvant therapy was administered, there was no significant difference in likelihood of undergoing surgical resection between White and non-White patients [40]. An NCDB study by Cloyd and colleagues that included 58,124 patients, patients with comorbidities, who were uninsured or had Medicaid coverage, or who lived in south or west geographic locations were more likely to receive surgery upfront and less likely to receive neoadjuvant chemotherapy for upfront resectable PDAC [41]. Additionally, patients were less likely to undergo resection if they lived in the Southeast region and were uninsured [42]. Yet, when patients did undergo resection, overall survival was not dependent on race or Social Deprivation Index, a measure of socioeconomic status [43,44]. Overall survival was worse for patients who were unemployed or lived in the South region of the United States [43].

In a systemic review encompassing the last twenty years, Fonseca and colleagues summarized the inequalities in access to cancer care for patients with PDAC [45]. They found that non-White patients were less likely to receive resection regardless of disease status (either localized disease or borderline resectable) or surgical referral in addition to being less likely to receive resection even if they had a surgical consultation [45]. Being from a lower socioeconomic status or being uninsured were negative predictors of receiving standard-of-care treatment for PDAC [45]. Access to care for PDAC is not only a function of patient factors but is also influenced by provider and system factors [45]. There is a critical need to address persistent disparities in access to cancer care for PDAC.

## 9. Summary 

Neoadjuvant chemotherapy for borderline resectable and locally advanced PDAC has been associated with an overall survival benefit. This is most likely due to downstaging of disease and greater likelihood of achieving negative surgical margins. The role of neoadjuvant radiation is currently unclear due to less favorable findings from the recent Alliance A021501 trial that demonstrated the addition of radiation therapy was associated with a lower proportion of patients proceeding to surgical resection and that those patients that did proceed to surgery were less likely to have an R0 resection. The benefit of adjuvant therapy for patients with borderline resectable/locally advanced disease who received neoadjuvant therapy is also uncertain, although current practice is to give adjuvant therapy to complete a total of a 6-month course. Overall, this review highlights the advances and remaining challenges in the treatment of borderline resectable/locally advanced PDAC.

## Figures and Tables

**Table 1 jcm-11-04866-t001:** Definitions of borderline resectable and locally advanced pancreatic cancer.

	Americas Hepato-Pancreato-Biliary Association/Society of Surgical Oncology/Society for Surgery of Alimentary Tract [3]	MD Anderson Cancer Center [4]	National Comprehensive Cancer Network [5]	Dana–Farber/Brigham Cancer Center
**Borderline Resectable**	Encase or abut SMV/PV	Encase the SMV/PV	Encase (>180-degree involvement) or abut (<180-degree involvement) the SMV/PV confluence	Abut, encase, or occlude a reconstructable portion of the SMV/PV
	Abut SMA, including encasement of either a short segment of the gastroduodenal artery or up to the hepatic artery		Contact with IVC	
			Abut SMA, without common hepatic artery involvement	
**Locally Advanced**	Any SMV/PV involvement not amenable to reconstruction or major venous thrombosis	Involve the celiac artery	SMV/PV involvement not amenable to reconstruction	Occlude a portion of the SMV/PV that cannot be reconstructed safely
			Encase the celiac artery or SMA	
	CHA involvement not amenable to reconstruction		Any degree of contact with aorta	
	Encase the SMA	Encase the SMA and have CHA involvement that is not amenable to reconstruction	Any degree of contact with the CHA	Encase the SMA, celiac trunk, or CHA
	Any involvement of the celiac artery		Extension to celiac axis or hepatic bifurcation	

Monroe Dunaway Anderson Cancer Center (MD Anderson Cancer Center), superior mesenteric vein/portal vein (SMV/PV), inferior vena cava (IVC), common hepatic artery (CHA).

**Table 2 jcm-11-04866-t002:** Summary of neoadjuvant chemotherapy.

	FOLFIRINOX	Gemcitabine/Nab-Paclitaxel
**Neoadjuvant Duration**	2 months	2 months
**Restaging**	After completion of 2 months neoadjuvant FOLFIRINOX	After completion of 2 months neoadjuvant gemcitabine/nab-paclitaxel
**Additional Neoadjuvant** **Duration**	2 months	2 months
**Surgical Resection**	If no disease progression and the patient is deemed surgically resectable, proceed to surgery	If no disease progression and the patient is deemed surgically resectable, proceed to surgery
**Adjuvant Duration**	2 months	2 months

## Data Availability

Not applicable.

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
