# Peer review of "Advances and Remaining Challenges in the Treatment for Borderline Resectable and Locally Advanced Pancreatic Ductal Adenocarcinoma"

_jcm, 2022, doi:10.3390/jcm11164866_

Round 1

Reviewer 1 Report

This manuscript presents potentially valuable review on recent advances in the neo-/adjuvant treatment of borderline resectable and locally advanced pancreatic carcinoma. 

Major comments:

A role of intraoperative (electron) radiotherapy (IOERT) should be discussed when evaluating adjuncts to surgery. 

Authors opinion on a selection of patients for neoadjuvant systemic alone or combined with the radiation therapy would be interesting. 

Reviewer 2 Report

I think this paper clearly describes the current situation and future directions of the treatment for  borderline resectable and locally advanced pancreatic ductal adenocarcinoma.

Minor revision:

I recommend that you include a table that summarizes neoadjuvant therapy.

Author Response

  1. I recommend that you include a table that summarizes neoadjuvant therapy.

Thank you to the referee for this comment. We have now included a Table 2 that summarizes neoadjuvant chemotherapy sequencing and duration. Additionally, we have described the chemotherapy regimen for FOLFIRINOX and Gemcitabine/Nab-Paclitaxel within the text, section 3 entitled “Neoadjuvant chemotherapy”. We state, “FOLFIRINOX regimen includes oxaliplatin (85mg/m2), irinotecan (180 mg/m2), and 5-fluorouracil (2400 mg/m2) administered every 2 weeks for a duration of 2 months. Subsequently, patients are restaged and then provided an additional 2 months of FOLFIRINOX therapy.[14] If patients do not demonstrate disease progression at restaging and are deemed candidates for surgical resection, they proceed to surgery. Alternatively, Gemcitabine/Nab-paclitaxel includes nab-paclitaxel (125 mg/m2) followed by gemcitabine (1000 mg/m2). This regimen is provided every week for 3 weeks followed by 1 week without chemotherapy for a total of 2 months (Table 2). Patients are then restaged and receive an additional 2 months of chemotherapy with plans to proceed to surgery if there is no disease progression and they are deemed candidates for surgical resection at restaging.”

The text is now added in line 111-121 and Table 2 is added in line 148.

Round 2

Reviewer 1 Report

I've found the revised version substantially improved. However, only because of the most recent publication of important relevant clinical data [JAMA Oncol. doi:10.1001/jamaoncol.2022.2319; Published online July 14, 2022.], I'd suggest to revise the manuscript with inclusion of this Reference. This would make your review article very attractive. 

Author Response

Thank you very much for the suggestion. We have revised this section in pages 6-7 and it now includes the following (citation 31 is now the recent publication by Katz that you referenced):

For patients who underwent a pancreatectomy, 88% (28 out of 32) of patients who had neoadjuvant modified FOLFIRINOX alone achieved an R0 resection compared to 74% (14 out of 19) of patients who had the addition of radiation therapy. [31] The two patients with a complete pathologic response were in the modified FOLFIRINOX plus radiation arm. Median overall survival was 29.8 months (95% CI 21.1 – 36.6 months) for the modified FOLFIRINOX alone arm versus 17.1 months (12.8-24.4 months) in the modified FOLFIRINOX with radiation arm. [31] Due to these findings, the authors concluded that the addition of radiation to modified FOLFIRINOX could be considered effective. [31]